# Prevalence of Primary Radiographic Signs of Hip Dysplasia in Dogs

**DOI:** 10.3390/ani12202788

**Published:** 2022-10-15

**Authors:** Stefania Pinna, Chiara Tassani, Alessandro Antonino, Aldo Vezzoni

**Affiliations:** 1Department of Veterinary Medical Sciences, University of Bologna, Via Tolara di Sopra 50, 40064 Ozzano dell’Emilia, Italy; 2Clinica Veterinaria Vezzoni, Via delle Vigne, 190, 26100 Cremona, Italy

**Keywords:** hip, dysplasia, radiography, FCI, dog

## Abstract

**Simple Summary:**

Hip dysplasia is the most common non-traumatic disease and cause of lameness in dogs. Different screening programs which use radiographic evaluation of the extended ventrodorsal projection of the hip exist for breeding selection. The Federation Cynologique International assigned five different scoring grades from A to E, ranging from a normal hip to severe dysplasia, to provide certification of the status of the joint. Of the different methods used for assessment, the Brass method is the most descriptive since it considers each part of the hip joint individually. The aim of the present study was to improve knowledge regarding the morphological changes in the context of hip dysplasia progression, especially in grades A to C. Some alterations were even found in grade A. The laxity and incongruence of the joint lead to the modification of the shape of the head and femoral neck, although some changes may be present in the normal or near normal hip joint. Some of the present findings, which were in contrast to the literature, are very interesting and additional investigation is indicated.

**Abstract:**

Hip dysplasia is a disabling orthopedic disease in dogs. The aim of this retrospective study was to evaluate the morphological aspects and radiographic changes in the development of hip dysplasia in adult dogs, and to describe the frequency of each radiographic parameter according to each Fédération Cynologique Internationale (FCI) grade. Brass descriptive forms for the evaluation of hip dysplasia were obtained from the archive of the Italian Fondazione Salute Animale, and the radiographic evaluation of 642 hips were processed. Sixteen radiographic criteria were assessed, divided into six main parameters: acetabulum, femoral head and its position in the acetabulum, femoral neck, joint space, and Norberg angle. The initial mild alterations were shown in the craniolateral acetabular rim (31.8%), slightly divergent in the joint space in 58.6% of FCI-A. The spherical shape of the femoral head was mildly small/flattened in 56.9% of FCI-B, in addition to a slightly cylindrical-shaped femoral neck (60.5%) and slightly lost contours (55.0%). Changes in acetabular depth (45.0%), and in the cranial acetabular margin (56.7%) were found in FCI-C. The center of the femoral head was lateral to the dorsal acetabular rim in 70.0% of FCI-B; the Norberg angle appeared normal in 70.6% of FCI-B. Elaboration of the radiographic criteria from the Brass descriptive forms allowed for the extrapolation of accurate knowledge regarding morphologic changes in the development of dysplasia by providing detailed information for each individual. In particular, the present survey showed that the morphological alterations of the acetabulum prevailed over those of the femoral head only at the beginning of the development of canine hip dysplasia, and then worsened after the changes occurred in the femoral head and neck.

## 1. Introduction

Canine hip dysplasia (CHD) is a multifactorial disease characterized by a malformation of the hip joints [1,2,3]. In 1966, CHD was described as laxity of the hip joint, leading to subluxation during early life and giving rise to varying degrees of flattening of the femoral head and acetabulum, consequently causing osteoarthrosis (OA) progression [4,5]. Since 1950, radiographic evaluation of the hip-extended ventrodorsal projection has commonly been used for the screening and diagnostic workup of hip dysplasia [3,6,7], even if other more accurate collateral diagnostic examinations have been described to examine details of the anatomy and laxity of the hip joint [8,9,10].

Radiographic study allows identifying primary and secondary alterations in the context of hip pathology. Passive laxity and lack of congruency are the first alterations detectable in the early life of dogs; however, they are not always indicative of the degree of hip dysplasia at skeletal maturity. Alterations in the shape and depth of the acetabulum, the shape of the femoral head and the femoral neck, and the size of the Norberg angle are primary signs; the biomechanical factors due to these bone changes could lead to chronic subluxation of the hip joint [2,4,5,9,11]. Secondary signs represent the hip joint degeneration consequent to laxity, incongruence, and inflammation, and are characterized by the formation of osteophytes and exostosis [5,12,13].

There are three internationally recognized CHD classification systems: the Fédération Cynologique Internationale (FCI) classification system, the Orthopedic Foundation for Animals (OFA), and the British Veterinary Association/The Kennel Club (BVA/KC) [5,14,15]. These are organizations applying score schemes used to describe the hip dysplasia status in dogs. The FCI system was utilized in the present study. The FCI assigns five different scoring grades, from A to E, ranging from a normal hip to severe dysplasia, respectively [16]. The grades are assigned based on primary and secondary alterations of the hip joint [3,12,17].

The Italian Kennel Club (ENCI), member of FCI, recognizes the Italian Fondazione Salute Animale (FSA) as the official panel certifying the status of hip dysplasia for all registered breeds, using three different methods to highlight the main morphologic aspects of the hip joint: the Professor Wilhelm Brass, the Professor Mark Flückiger, and the Professor Malcolm B. Willis methods.

The Prof. M. Flückiger and Prof. M. B. Willis methods consist of numerical schemes obtained from evaluating the various radiographic aspects of Swiss and English breeds, respectively. The total score provides the hip dysplasia grading.

Professor Brass’ method consists of a descriptive scheme of the various aspects of the hip joint and of the alterations induced by mild or severe dysplasia, for any breed registered by the FCI. The Brass method was utilized in this study since it is an analytic description of all the aspects of the hip joint modifications, leading to the final score according to the FCI description of the five degrees of hip dysplasia, as reported in Table 1. The quality of the radiographic images and the hip and femur positioning were assessed, since correct positioning is required for identifying minimal deviations from anatomical conformation [5]. The radiographic parameters related to the joint, such as the acetabulum, femoral head and neck, femoral head position within the acetabulum, joint space, and Norberg angle, were then evaluated.

Several aspects of each of these parameters were studied and were assessed as normal, mild alteration or severe alteration [16,18].

Each hip joint was considered individually since the alterations present in one hip joint could be different from those in the other hip joint. The final evaluation of the general status of the dog was based on the determination of the most compromised joint [12,19].

The primary and secondary alterations which characterize hip dysplasia in dogs have been well described [11,15,16,20,21]. However, to the authors’ knowledge, the prevalence of one alteration over another during the development of hip joint dysplasia in adult dogs has never been described.

In this study, the primary aim was to evaluate the morphological aspects and radiographic changes in the development of CHD according to the Brass descriptive forms, investigating the most frequent alterations present in the various FCI grades and the first to appear in the CHD. An additional goal was to describe the frequency of each radiographic parameter within each FCI grade.

## 2. Materials and Methods

The study was conducted by processing the technical evaluation forms previously completed and graded for hip dysplasia by at least two ECVS and ECVDI diplomates. The forms on which the dysplasia score was recorded for each registered dog were obtained from the Italian FSA archive, and were approved and certified by the FCI. The FSA had received the radiographs taken from FSA-licensed veterinary radiology laboratories located throughout the country to be evaluated and certified. The form is a technical evaluation of CHD and consists of three parts corresponding to the three methods: the Prof. W. Brass, the Prof. M. Flückiger, and the Prof. M.B. Willis methods. All the forms were completed using the Brass method for all dogs; in addition, the Swiss breeds also required evaluation using the Flückiger method and the English breeds using the Willis method, according to the BVA/KC method.

The database from the FSA was searched, and the hips of dogs which had been scored at 12 months or older using the FCI system were randomly selected. A total of 321 technical evaluation forms regarding CHD were selected for the study; each hip joint was considered individually. Only pure breeds were included: Labrador Retriever (*n* = 90), Golden Retriever (*n* = 80), German Shepherd dog (*n* = 59), Bernese Mountain dog (*n* = 35), Rottweiler (*n* = 13), Border Collie (*n* = 7), Czechoslovakian Wolfdog (*n* = 6), Australian Shepherd (*n* = 5), Lagotto dog (*n* = 4), Greater Swiss Mountain dog (*n* = 4), Weimaraner (*n* = 3), Rhodesian Ridgeback (*n* = 3), Doberman (*n* = 3), Pumi (*n* = 2), Cane Corso dog (*n* = 2), White Newfoundland dog (*n* = 1), Black Newfoundland dog (*n* = 1), English Setter (*n* = 1), American Staffordshire terrier (*n* = 1), and Miniature Schnauzer (*n* = 1).

The criteria for inclusion of the forms were those corresponding to hips certified according to the FCI grades FCI-A, FCI-B, FCI-C or FCI-D. The forms had to be completed using the Brass method.

Forms completed for FCI-E were excluded, since degenerative changes of hip dysplasia are already advanced and evident at that stage. The goal of this study was the evaluation of the primary signs which are obscured by moderate to severe OA. The parts of the forms completed using the Willis and Flückiger methods were not processed since they had been completed only for a limited number of breeds. Five forms for breeds represented by a single individual were also removed from the study.

The criteria of the Brass descriptive method include the evaluation of the acetabulum (namely acetabular depth, cranial acetabular margin, craniolateral acetabular rim, and acetabular osteophytes), the femoral head, the position of the femoral head in the acetabulum, the femoral neck, the joint space, and the Norberg angle measurement.

The radiographic parameters were evaluated as normal, mild or severe alterations for each hip, and each of these parameters arbitrarily received an identification score of 0, 1, or 2, respectively. In Table 2, the criteria of the Brass descriptive method are listed in columns and their evaluation in rows in order of increasing severity.

Acetabulum. The acetabular depth (AD) indicates the acetabular coverage of the femoral head; it is represented by the distance between the dorsal acetabular rim (DAR) and the medial acetabular wall. This parameter depends on the shape of the acetabulum; it is independent of the femoral head position [20,22]. When the acetabulum is flatter than normal, the distance decreases. The acetabulum can be classified as normal, or mildly, or severely flattened (scores 0, 1, and 2, respectively).

The subchondral bone density of the cranial acetabular margin (CAM) is an easily detectable line between the acetabular notch and the dorsal acetabular rim [20]. It is usually fine and uniform (score 0). When the subchondral bone density becomes sclerotic, the margin becomes less sharp, and its radiopacity increases (scores 1 and 2, respectively).

The craniolateral acetabular rim (CLAR) is the point in which the dorsal acetabular rim curves around to become the cranial acetabular margin. It is slightly curved, and is considered normal when its lateral extension follows the curvature of the femoral head (score 0) [20]. The CLAR can lose this curvature and become mildly or severely flattened (scores 1 and 2, respectively).

Acetabular osteophytes (AOsts) may be visible along the capsular insertion, as mild irregular proliferations along the CLAR (score 1), up to a bilabiation (score 2) [23]. Score 0 was not applicable.

Femoral head. The femoral head (FH) normally has a spherical shape (SFH) with a small depression corresponding to the fovea capitis (score 0). When shape alteration is present, the femoral head may be slightly small or flattened (score 1), or obviously small or flattened (score 2).

Collar in the femoral head (CFH). Collar exostosis is a formation along the base of the head at the cranial insertion of the joint capsule. Collar exostosis, from slight to obvious, may be observed (scores 1 and 2). Score 0 was not applicable. A deformed femoral head (DFH) is characterized by exostosis, having a mushroom-like appearance [24]. The alteration in shape may be mild (score 1) or deformed (score 2) whereas score 0 was not applicable.

Position of the femoral head in the acetabulum. The femoral head is deeply seated within the acetabulum (PFH) in a normal hip (depth: score 0); otherwise, a degree of subluxation may be detected: slightly loose or loose (scores 1 and 2). Moreover, the center of the femoral head must be medial to or superimposed on the line drawn along the dorsal acetabular rim (FHC/DAR). In the case of subluxation, the center is positioned lateral to the radiographic line of the DAR: center lateral (score 1) and center markedly lateral (score 2).

Femoral neck. The femoral neck (FN) is considered to be normal when it becomes thin (TFN) towards its center (score 0) and when it is clearly identifiable (IFN) relative to the femoral head (score 0) [20,24]. Otherwise, a TFN could look like the extension of the femoral head due to its slightly cylindrical shape or cylindrical shape (scores 1 and 2). Scores 1 and 2 were not applicable for the IFN parameter. The femoral neck normally has (score 0) sharp contours (contour of femoral neck, CFN). The loss of sharp contours may be slight or obvious (scores 1 and 2), due to osteophytes (FNOst) and the Morgan line (MoFN) which represent signs of OA progression [25]. For the FNOst and MoFN alterations, score 0 was obviously not applicable. A Morgan line could be present in large size, active dogs as an enthesophyte at the caudal insertion of the joint capsule; when it is thin and not associated with other changes, it is not a sign of CHD [26,27].

The joint space (JS) is regular when the circle drawn on the femoral head is inside the circle drawn on the cortical edge of the acetabulum. It is considered mildly to severely altered when the circles are not concentric, and the joint space between the femoral head and the acetabulum is irregular [20]. The JS was considered to be concentric, slightly divergent and divergent, corresponding to scores 0, 1, and 2, respectively.

The Norberg angle (NA) is the principal measurement quantifying the degree of femoral subluxation. The NA is represented by the angle between a line which connects the center of the femoral heads and a line which connects each femoral head with the lateral point in which the cranial acetabular margin meets the dorsal acetabula margin, called the craniolateral effective rim [4,6]. An NA of approximately 105° defines good acetabulum coverage and a congruent hip, and is considered to be normal [28]. An NA less than 105° could be indicative of hip laxity and of divergence of the hip joint space [6]. The following scores were assigned to the NA measurement of the Brass form: score 0 (> 105°) for a normal hip, score 1 (<105°) and score 2 (< 100° or < 90°) for increasing severity of the subluxation.

Statistical Analysis. Data regarding breed, age, body weight, sex, FCI, and grade of dysplasia were collected. The radiographic criteria obtained from the Brass form, described above, were also collected. The continuous data were evaluated using the Kolmogorov–Smirnov test for normal distribution. The continuous data were reported as medians, and ranges (minimum and maximum values), and 95% CI. The categorical data were evaluated as frequencies and/or percentages. Labrador Retrievers, Golden Retrievers, German Shepherd dogs and Bernese Mountain dogs were assessed since they were the most common breeds in the database.

All the data were elaborated as descriptive statistics using the statistical software program MedCalcR Software 16.8.4 (Ostend, Belgium).

## 3. Results

The study population included 316 dogs, belonging to several breeds. There were 152 males (48.1%) and 164 females (51.9%). The median age was 14 mo (range: 12–96 mo, 95% CI: 13–15 mo), and the median weight was 32 kg (range: 12–57 kg, 95% CI: 31–32 kg). Based on the FCI grades, the dogs were assigned to FCI-A (*n* = 162, 51.3%), FCI-B (*n* = 109, 34.5%), FCI-C (*n* = 30, 9.5%) and FCI-D (*n* = 15, 4.7%). Descriptive data regarding age and weight in each FCI grade are shown in Table 3.

Labrador Retrievers, Golden Retrievers, German Shepherd dogs and Bernese Mountain dogs were described according to age, weight and FCI grade. There was a prevalence percentage of Labrador Retrievers and Golden Retrievers of 28.5% and 25.3% of the population, respectively. The distribution of FCI grades showed the greatest percentage of the FCI-A group to be Labrador Retrievers at 70% and Bernese Mountain dogs at 54.3% of the sample breeds examined. All the other results are listed in Table 4 and Table 5.

A total of 632 joints were studied. Based on the FCI grades, the hips were assigned to FCI-A (*n* = 324), FCI-B (*n* = 218), FCI-C (*n* = 60), and FCI-D (*n* = 30). Each parameter reported on the FCI form regarding the entire population was elaborated (according to the Brass descriptive method), and the scores 0, 1 and 2 were counted and reported in percentage values for each of the four FCI groups. Only the most significant results are reported below while the total data are shown in Figure 1.

Acetabulum. Three parameters were evaluated for the acetabulum: acetabular depth (AD), cranial acetabular margin (CAM) and craniolateral acetabular rim (CLAR). Acetabular osteophytes (AOsts) were also reported. In the FCI-A group, the acetabulum was normal (score 0) for all parameters (AD 100%, CAM 97.2%, CLAR 68.2%); however, 31.8% of the hips showed the CLAR with mild alteration (score 1).

In the FCI-B group, the number of dogs having a score of 0 for each parameter decreased (AD 85.3%, CAM 72.5%, CLAR 22.0%), combined with a marked increase in score 1 for the CLAR (78.0%).

The FCI-C group showed an increase in the percentage of score 1 (AD 45.0%; CAM 56.7%, CLAR 43.3%), and of score 2 (CAM 10.0%, CLAR 38.3%). In this group, the first sign of acetabular osteophytes (AOst 20.0% with score 1, and 6.7% with score 2) appeared.

In the FCI-D group, the acetabulum showed an increase in the scores of all the parameters; however, score 0 was still present in 40% of the AD scores and in 16.7% of the CAM scores.

Femoral Head (FH). The shape of the FH was radiographically evaluated as follows. In the FCI-A group, the head was normal (SFH 98.1%) without any alterations. In the FCI-B group, the percentage of score 1 for spherical shape of the FH increased and the presence of collar exostosis was recorded (SFH 56.9%, CFH 2.3%). In the FCI-C group, there was a worsening, with an increase in score 1 (SFH 80.0%, CFH 41.7%), and score 2 (SFH 5.0%, CFH 20.0%). The FCI-D group showed obvious modification of the SFH with an increase in both score 1 and score 2 (SFH 60% and 23.3%, respectively), and of the DFH in score 1 (DFH 13.3%).

Position of the femoral head in the acetabulum. Both the PFH and the FHC/DAR showed a similar tendency within each FCI grade; both had a similar prevalence of score 0 in the FCI-A group, and of score 1 in the FCI-B group, and also of score 2 in the FCI-C group with a percentage of 45.0% for the FH/DAR versus 26.7% for the PFH. In the FCI-D group, there was a high prevalence of score 2 (63.3%) for both parameters.

Femoral Neck (FN). The FCI-A group had a normal FN with the TFN, IFN, and CFN scoring 0 (94.1%, 96.0%, and 97.2%, respectively).

In the FCI-B group, there was an increase in score 1 for the TFN, and CFN (60.5% and 55.0%, respectively), with a ratio of almost 1:1 for score 0 and score 1. No OA signs were recorded.

In the FCI-C group, 15.0% of the TFNs had score 2; however, 26.7% of the hips still had a normal FH (TFN score 0). Osteophytes (FNOst score 1) were observed in 11.7% of the hips.

In the FCI-D group, all the parameters showed an obvious increase in score 2.

Joint space (JS). The JS was found to be slightly divergent in 58.6% of the hips (score 1) in the FCI-A group, 97.3% in the FCI-B group, and appeared divergent (score 2) in 31.7% of the FCI-C group. In the FCI-D group, there was a prevalence of score 2 of 63.3%.

Norberg angle (NA). The NA was slow to change: 99.4%, 70.6% and 43.3% for score 0 in the FCI-A, -B, and -C groups, respectively. Score 2 was assessed in 21.7% of the FCI-C hips and in 53.3% of the FCI-D hips.

## 4. Discussion

Elaboration of the radiographic criteria of CHD taken from the technical evaluation forms allowed for the extrapolation of clear and accurate facts regarding the morphologic changes in the development of CHD from grades A to C of the FCI scoring system. In clinical practice, this knowledge could be useful in the decision-making process regarding early surgical options for preserving the hip joint. The main purpose of the present study was to retrospectively evaluate the morphological aspects and radiographic changes in the development of hip dysplasia in adult dogs. Analysis of 316 technical evaluation forms of hip dysplasia led to the elaboration of 632 hips having an accredited FCI score. Currently, the FCI scoring mode for grading CHD describes radiological features, mainly concerning the congruence between the femoral head and the acetabulum, and the measurement of the Norberg angle. The craniolateral acetabular rim and the signs of osteoarthrosis were also described [16,18,29]. Despite this, the attribution of a CHD grade was the result of evaluating a set of numerous parameters which could be quantified differently within the same grade, and between the two sides of the hip. The technical evaluation form of hip dysplasia provides a descriptive assessment of the radiographic changes of the hip, according to the Brass method. Therefore, in the present study, the Brass forms were elaborated in order to obtain an additional detailed description of the changes for each radiographic parameter.

Pathological changes in the acetabulum, and the femoral head and neck have been well described in the literature, [2,9,11,20,24,30]; however, they have not been reported simultaneously and, therefore, an immediate comparison of one prevalence over the other was lacking. The results of the present survey showed that the morphological alterations of the acetabulum had a prevalence over those of the femoral head only at the beginning of the development of the CHD. The CLAR was already mildly flattened in 31.8% of the normal hips (FCI-A), which was confirmed by a slight divergence of the JS (58.6%). The CLAR alteration increased markedly in the FCI-B group by 46.2 percentage points. The first change in the sphericity of the femoral head (56.9%) was recorded in the FCI-B group. The other two radiographic parameters for the acetabulum (AD and CAM) showed worsening at a later time.

This tendency could be both a consequence and a cause of the biomechanical alterations of CHD. In a healthy and congruent hip joint, forces during weight bearing are distributed across the entire cartilaginous surface of the acetabulum [31,32,33,34]. If the conformation, and therefore the stability, of the hip joint is compromised, abnormal biomechanical forces start to wear out the structures of the acetabulum, and the femoral head and neck. The loading patterns of canine hip joints suggested that the initial main forces were concentrated on the superior end of the femoral head and then spread medially [34]. When there is instability and incongruence, the forces are concentrated in a small contact area corresponding to the CLAR [31,33]. Wolff’s law states that “every change in the form and function of a bone, or of function alone, is followed by certain definite changes in the internal architecture and equally definite secondary alterations in the external conformation in accordance with mathematical laws” [34]. Following this law, the acetabulum progressively flattens, and becomes unable to seat the already modified head; this was confirmed by the position of the head. The PFH was slightly loose, and the FHC/DAR had the center of the head lateral to the DAR; both had a prevalence of score 1 in the FCI-B group which then increased to score 2 with a loosening of the PFH; the FHC/DAR center was markedly lateral in the FCI-C group.

In agreement with the literature, articular incongruence or divergence of the joint rim is an early sign of CHD. The FCI guidelines assess the center of the femoral head placed medially or superimposed on the DAR as a normal position, and laterally and markedly lateral when the congruence is lost. It would be interesting to differentiate the first two positions, medial and superimposed, as cut-offs between normal or near normal hip joints. The present report was a retrospective study; therefore, it was not possible to extrapolate these exact data from the Brass descriptive forms.

Unexpectedly, the early alterations of the CLAR did not match with the flattening of the AD, which was normal in the FCI-B group, with slight subchondral sclerosis of the CAM; the depth decreased only mildly in the FCI-C group. An early lack of development of the CAM is described in the literature, and appears as a flattened and decreased AD, followed by a remodeling of the femoral head and neck [20].

In agreement with the literature, the subluxation of the femoral head is usually the earliest radiographic sign of CHD [7,16,20]. In the present study, the JS was slightly divergent in the FCI-B group; however, the NA did not decrease and was prevalently > 105°, as the NA changes more slowly than the JS. This behavior could have been due to the difficulty in finding the correct landmark in the craniolateral effective rim, whereas the FHC/DAR is easier to visualize and, when supported by the JS, more importance could be given to the attribution of the dysplasia score.

The results of this study revealed that the shape of the femoral head was already mildly flattened in the FCI-B group, and worsened in the FCI-C group along with the appearance of the first signs of collar exostoses. Assessment of the acetabular parameters, such as AD and CAM, revealed that a normal score was still prevalent in the FCI-B and -C grades of CHD progression.

The femoral neck showed changes concomitant with those of the head, even in the FCI-C group in which the slight loss of sharp contours, and the appearance of osteophytes and Morgan’s line were associated with a slight loss of sphericity and collar formations, however without deformity of the head. Unexpectedly, the results of this study showed that the shape of the head and neck was already changed in the hip joints classified as near normal or as mild dysplasia (FCI-B and FCI-C, respectively). The fact is that any skeletal alteration is a sign of the development of dysplasia and should be reconsidered when attributing the severity of the dysplasia scores.

Several authors have described the development of hip dysplasia, and some differences were revealed. Slocum described the radiographic characteristics of hip dysplasia with the following sequence of appearances: incongruence, Morgan’s line, thickened neck, and decreased acetabular depth. Deformities of the head appeared subsequently while osteophytes were already visible in the early stage of the dysplasia [35]. In 1992, Henry described the radiographic development of the healthy hip as well the dysplastic hip during the growth of the dog from birth up to 36 months of age. He stated, in accordance with the literature, that subluxation was considered by many to be definitive evidence of the presence of CHD. The description of pathological radiographic changes in adult dogs concerned both the acetabulum, and the head and neck of the femur, which were explained separately [20].

The FSA form used in the present study includes three evaluation methods. Prof. M. Flückiger’s method is used only for Swiss breeds and Prof. M. B. Willis’ method, according to the BVA/KC method, evaluates English breeds. Prof. W. Brass’ method is used for all breeds, although the Swiss and English breeds were also evaluated by the other method in the same form [15,17]; (https://www.fondazionesaluteanimale.it, last accessed on 15 June 2022). Therefore, the choice of the Brass method to conduct this survey was justified by the sample with the largest number of breeds which simulates a population of dogs. In addition, some differences were noted between the three methods regarding CHD scoring; this difference excluded the first two methods from this investigation. The Swiss scoring mode, according to Prof. Fluckinger, evaluates six radiographic parameters, using five scores for each one, and provides five grades of dysplasia for each hip joint [17]. Brass evaluates six main parameters for a total of sixteen radiographic criteria. Unlike Prof. Brass’ form, the Swiss method evaluates some parameters together as if they were one, such as the position of the center of the femoral head relative to the dorsal acetabular rim assessed together with the width of the joint space, or the shape of the femoral head evaluated with the femoral neck. The femoral neck is assessed for the extension of exostosis; the Brass form, on the other hand, provides five parameters for reporting the changes in the femoral neck. The BVA/KC score system evaluates the head and neck as the only parameter [17]. The investigation carried out using the Brass method revealed that the femoral head and neck changed simultaneously, providing a precise and detailed description. The descriptive scheme of the Brass method, rather than the scoring scheme of the other two, made Brass more suitable for the purpose of this study. However, it would be interesting to investigate a sample of hip joints evaluated with all three methods; however, the specificity of the Swiss and English breeds does not allow the simultaneous compilation of all the three methods.

With regard to the above, the Brass method does not give a numerical score as do the Swiss and the BVA/KC methods; however, it describes several features of the hip joints and shows detailed morphological changes to be taken into account when scoring. This method can be considered to be a reminder of the various alterations existing in a hip joint, which can be found in the characteristics listed for each FCI grade. Scores from 0 to 2 were adopted by the present study purely to obtain a sequence of the occurrence of the skeletal alteration; they did not have a numerical value.

It is useful to indicate the age of the dogs at the time of the HD evaluation, as advancing age can exacerbate secondary radiographic appearances, i.e., increased signs of OA. The median age was 14 months, with a CI of 13–15 months. The study did not include younger animals because they would not be eligible for official evaluation by international organizations. In the present study, only a few dogs were over three years of age, since the certification objective was mainly related to selection for canine breeding, and this number was not sufficient for making a substantial comparison. However, this was in accordance with the goal of the present study which was to assess the prevalence of primary radiographic signs of HD in dogs.

In the present study some limitations should be noted. One limitation of the study was that the alterations of FCI grade D were not elaborated due to the small number of samples in which all the parameters underwent severe changes. Since the Brass form evaluates the two hips separately, additional research would be interesting in order to report the differences in the same dog. In the present study, the better hip could have biased the results within an FCI grade. Another limitation could have been breed interference; in other words, breed-specific conformation can influence biomechanical differences. Despite this, the FCI considers the same criteria for evaluating hips and degrees of dysplasia equally for all breeds, taking into account only the morphological differences in the chondrodystrophic breeds for the shape of the femoral head. In the present study, the number of dogs of the four most common breeds was limited, in particular when grouped into FCI grades, and was therefore not significant for a comparison of the radiographic criteria with each other. However, the analysis of the four most common breeds found in the present study revealed that Labrador Retrievers, Golden Retrievers and Bernese Mountain dogs were prevalently classified as FCI-A, and German Shepherd dogs were prevalently classified as FCI-B. No differences were noted in median age and weight to explain the prevalence of these breeds, although Bernese Mountain dogs are certified after 15 months of age, and have a heavier median weight. In the near future, it could be interesting to elaborate each radiographic parameter according to breed; however, in the present study, as aforementioned, the small number of dogs obtained with these groupings did not allow it.

## 5. Conclusions

In accordance with the pathogenesis of CHD, the skeletal alterations found in this study began in the acetabulum, but only in the craniolateral acetabular rim. The radiographic changes evolved with prevalent severity of the head and femoral neck, already evident in the FCI-B group. The acetabulum was of normal depth in half of the joints investigated in the FCI-C group.

Some of the present findings which were in contrast to the literature are very interesting and additional investigation is indicated. In particular, a future survey between homogenous groups regarding age, weight and breed would be of great value in understanding the development of canine hip dysplasia.

## Figures and Tables

**Figure 1 animals-12-02788-f001:**
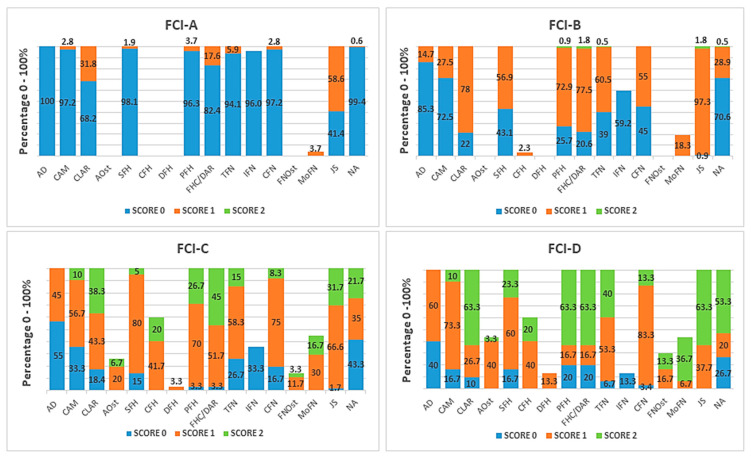
The charts show the percentage of the scores (0–2) of the radiographic parameters of hip dysplasia according to Prof. Brass’ form. Grades A, B, C and D indicate the FCI classification. Abbreviations are listed in Table 2.

**Table 1 animals-12-02788-t001:** The 5-grade FCI classification of canine hip dysplasia (Dortmund 1991, updated 2015).

**Grade A**	**No signs of hip dysplasia** The femoral head and the acetabulum are congruent and the acetabular angle according to Norberg is about 105° (as a reference). The joint space is narrow and even. The craniolateral rim appears well defined and slightly rounded. There are no signs of osteoarthrosis.
**Grade B**	**Near normal hip joints** The femoral head and the acetabulum are slightly incongruent and the acetabular angle according to Norberg is about 105° or, the femoral head and the acetabulum are congruent and the Norberg angle is about 100°. The center of the femoral head lies medial to the dorsal rim of the acetabulum. There are no signs of osteoarthrosis.
**Grade C**	**Mild hip dysplasia** The femoral head and the acetabulum are incongruent, the acetabular angle according to Norberg is about 100° and/or there is a slightly flattened craniolateral rim. Irregularities or no more than slight signs of osteoarthritic changes in the margo acetabularis, cranialis, caudalis or dorsalis, or in the femoral head and neck, may be present.
**Grade D**	**Moderate hip dysplasia** Obvious incongruency between the femoral head and the acetabulum with subluxation. Acetabular angle according to Norberg more than 90° (only as a reference). Flattening of the craniolateral rim and/or osteoarthrotic signs.
**Grade E**	**Severe hip dysplasia** Marked dysplastic changes of the hip joints, such as luxation or distinct subluxation, acetabular angle according to Norberg less than 90°, obvious flattening of the margo acetabularis cranialis, deformation of the femoral head (mushroom shaped, flattening) or other signs of osteoarthrosis.

**Table 2 animals-12-02788-t002:** Radiographic parameters, abbreviations, and scores according to Prof. Brass’ form.

Radiographic Parameters	Abbreviations	Score = 0	Score = 1	Score = 2
**Acetabulum**				
Acetabular depth	**AD**	Normal depth	Mildly flattened	Severely flattened
Cranial acetabular margin	**CAM**	Normal line	Slight subchondral sclerosis	Obvious subchondral sclerosis
Craniolateral acetabular rim	**CLAR**	Slightly curved	Mildly flattened	Severely flattened
Acetabular osteophytes	**AOst**	na	Mild	Severe
**Femoral Head (FH)**				
Spherical Femoral Head	**SFH**	Spherical shape	Mildly small/flattened	Small/flattened
Collar Femoral Head	**CFH**	na	Some collar exostosis	Obvious collar exostosis
Deformed Femoral Head	**DFH**	na	Slightly deformed	Deformed
**Position of the femoral head in the acetabulum**				
Position of the Femoral Head	**PFH**	Depth	Slightly loose	Loose
Femoral Head Center/Dorsal Acetabular Rim	**FHC/DAR**	Center medial or superimposed	Center lateral	Center markedly lateral
**Femoral Neck (FN)**				
Thin Femoral Neck	**TFN**	Thin	Slightly cylindrical shape	Cylindrical shape
Identifiable Femoral Neck	**IFN**	Identifiable	na	na
Contours of the Femoral Neck	**CFN**	Sharp contours	Slightly lost contours	Lost contours
Femoral Neck Osteophytes	**FNOst**	na	Some osteophytes	Severe osteophytes
Morgan Femoral Neck	**MoFN**	na	Slight Morgan line	Obvious Morgan line
**Joint Space**	**JS**	Concentric	Slightly divergent	Divergent
**Norberg Angle**	**NA**	>105°	<105°	<100° and <90°

Legend: na, not applicable.

**Table 3 animals-12-02788-t003:** Distribution of the data of the total population in sex, age, weight, and FCI-grade.

FCI Grade	*n*. Dogs (%)	Sex		Age (mo)			Weight (kg)		
		*n*. (%) M	*n*. (%) F	Median	Range (Min-Max)	95% CI	Median	Range (Min-Max)	95% CI
FCI-A	162 (51.3)	83 (51.2)	79 (48.8)	13.5	12–96	12–15	32	12–57	31–33
FCI-B	109 (34.5)	50 (45.9)	59 (54.1)	13	12–78	12–15	31	13–56	30–32
FCI-C	30 (9.5)	14 (46.7)	16 (53.3)	14.5	12–47	12–18.8	32	13–51	28.2–34.8
FCI-D	15 (4.8)	5 (33.3)	10 (66.7)	15	12–34	12–21	30	25–37	29.3–33.7

Legend: M, male; F, female.

**Table 4 animals-12-02788-t004:** Distribution of the data of the four breeds regarding sex, age and weight.

Breed	*n*. Dogs (%)	Sex		Age (mo)			Weight (kg)		
		*n*. (%) M	*n*. (%) F	Median	Range (Min-Max)	95% CI	Median	Range (Min-Max)	95% CI
LR	90 (28.5)	51 (56.7)	39 (43.3)	12	12–68	12–14	30	25–45	30–31.7
GR	80 (25.3)	50 (45.9)	59 (54.1)	12	12–96	12–14	31	25–44	30–32
GSd	59 (18.7)	14 (46.7)	16 (53.3)	12	12–48	12–14	32	22–50	30–33.5
BMd	35 (11.1)	5 (33.3)	10 (66.7)	16	15–48	15–23	40	28–57	36.2–45

Legend: LR, Labrador Retriever; GR, Golden Retriever; GSd, German Shepherd dog; BMd, Bernese Mountain dog.

**Table 5 animals-12-02788-t005:** Distribution of the data of the four breeds regarding FCI-grade.

FCI Grade	Breed—*n*. Dogs (%)			
	LR	GR	GSd	BMd
FCI-A	63 (70.0)	41 (51.3)	19 (32.2)	19 (54.3)
FCI-B	17 (18.9)	27 (33.8)	31 (52.5)	10 (28.6)
FCI-C	7 (7.8)	4 (5.0)	8 (13.6)	4 (11.4)
FCI-D	3 (3.3)	8 (10)	1 (1.7)	2 (5,7)

Legend: LR, Labrador Retriever; GR, Golden Retriever; GSd, German Shepherd dog; BMd, Bernese Mountain dog.

## Data Availability

The data presented in this study are available on request from the corresponding author. The data are not publicly available as the data was acquired from a third-party source.

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
