# Peer review of "Prevalence of Primary Radiographic Signs of Hip Dysplasia in Dogs"

_animals, 2022, doi:10.3390/ani12202788_

Round 1

Reviewer 1 Report

Comments to the authors

Thank you for submitting an interesting article. My comments are as follows.

l  This paper concludes that skeletal changes in CHD begin at the acetabulum when evaluated using the FCI classification. If this conclusion is correct, I offer the following opinions. 

[Abstract] This conclusion should be stated in the Abstract.

[Research design] It is necessary to confirm whether the other two classification methods fail to detect acetabular changes at an early stage. Without a comparative study, the FCI classification cannot be considered superior.

[Discussion ] 

・Please explain what features of the FCI classification lead to early detection of acetabular changes.

・Please explain what other two evaluation methods are suitable for detecting changes in CHD.

Author Response

Dear Reviewer, thank you very much for your comments. We are pleased to know that you found our research interesting and we would like to thank you for your suggestions for improving the quality of the study.

Reviewer 2 Report

Comments to the authors

16     If  alterations were found in Grade A – means that the scrutineers din`t consider the definitions exactly.

18     changes – pathological or normal variation?

30     a cylindric formed FN is only pathologic if caused by additional bone. In many dogs this form is visible in postion 1 (FCI) but in position 2 a normal TFN is presented.

                   INFLUENCE OF FEMORAL HEAD AND NECK CONFORMATION ON HIP

DYSPLASIA IN THE GERMAN SHEPHERD DOG

ANTJE WIGGER, BERND TELLHELM, MARTIN KRAMER, HEIKE RUDORF

                   Veterinary Radiology & Ultrasound, Vol. 49, No. 3, 2008, pp 243–248.

97      Materials and Methods

         It is not mentioned if the HD examinations have been performed by only one or by different scrutineers.

Table 2

         This are very subjective criteria for the different scores. Especially if different scrutineers are included. E. g.:

         CLAR: very often it is horizontal. This variation is not graded.

CAM: thickness and density can vary normally depending on breed and use.

         SFH: definition of „small“??

155   The FHC can be lat without subluxation and not any subluxation leads to a lat postion of FHC.

161   FN – see line 30

182   NA in the FCI definition is „about“ a degree. Here the NA is used as a treshold to score HD A.

220   CLAR: the obvious increase of „mildly flattened“ form hast o be discussed. Per FCI definition this form leads to HD „C“.

270   „.. FCI grade „E“ – not D…

302   head. . The PFH not PFN

306   …divergence of the joint rim ….   Means „joint space“?

333 – 337             In general the use of subjective, not exactly to define criteria and possibly different scrutineers is not included in the discussion. Also not the fact of normal anatomic variations. The use of MoFN and CFH as criteria is questionable.

The results of this study can only refered to this material and this scrutineer(s).

It would be helpful to show examples for the criteria.

That means not, that the study gives no valuable information. But it hast o be included in the discussion.

One important result seems to  that that the strict application of the FCI definitions is not possible or at least not reasonable to classify the HD grades.

Author Response

Dear Reviewer, I thank you for the additional suggestions which will surely improve the understanding of my study. I read your comments and noted that you have a different timing of correction than the other reviewers. You reviewed the first manuscript (June 20), that is, not the one already updated with the suggestions of the other three reviewers (August 26). Perhaps you might find my study already improved, as the others have appreciated. I will try to follow your suggestions on this latest manuscript, even if the line numbers do not match.

Reviewer 3 Report

The subject of hip dysplasia in dogs is always worth research and scientific analysis. Therefore, with great interest and enthusiasm, I welcomed the article for review.
After reading it, some basic questions arise:
- Have the authors seen the radiographs of the analyzed hip joints? Did they just rely on the completed forms?
- From how many veterinary radiology laboratories did randomly selected data come from? - how many veterinary radiologists described the radiographs selected for the research group? - one or many? As you know, there may be one radiograph and several different interpretations of the changes. Radiological evaluation is very subjective and depends on many factors, e.g. the patient's position for radiological examination. A slight asymmetry in the position is enough and the Norberg angle measurements are distorted.
- why were the dogs not divided into weight groups? After all, it is known that body weight is one of the primary factors in the occurrence of CHD.
- the changes in the Miniature Schnauzer (12 kg) and the Newfoundland (62 kg) cannot be compared in this scoring system!
- why is it not divided into age groups? The type and intensity of changes are spread over time and depend, for example, on the intensity of use.
- why were "some British and Swiss races" treated differently? This introduces some chaos and reduces the reliability of the results obtained!
- the statistics are very poor. One may be tempted to calculate the significance for individual parameters.
- conclusions are subject to redrafting because they are a generalization and contribute little.

Author Response

Dear Reviewer, I would like to thank you for your very helpful advice. I have modified the manuscript following all your suggestions. I carried out a careful and thorough review of the manuscript and tried to make it clearer. I hope I have resolved all your doubts.

Reviewer 4 Report

Interesting article, interesting approach to the issue, but I have some comments, which I present below.

In chapter 2. Materials and Methods did not mention exactly what breeds of dogs and how many individuals in the breed were tested. However, this information can be found in the next chapter, Results. Please, move this fragment of rows 192-199) to the appropriate place for the Material and methods chapter.

Moreover, in each breed, the distribution of sex, age and body weight should be provided, especially since a single individual was tested in 5 breeds. The same applies to the division within each breed into individual classes from Ado D (E not found in the present study). Combining the results for the Newfoundland and the English Setter is not allowed. Due to such a wide variety of breeds, it cannot be treated as Canis lupus familiaris. The difference between domestic dog breeds is not only the difference in body weight, but also in the massiveness of the bones, angle of the limbs, etc.

I believe that the article needs to be thoroughly rewritten, including the performance of analyzes broken down by breeds, and single samples, absolutely removed from the research. The authors themselves do not refer to breeds in the chapter results, which is perhaps the most interesting. The research is done so that they have some application, not the so-called "art for art". Throughout the work, no mention was made of the equipment performing specific analyzes and how many repetitions of these tests were performed.

Row 352 - reference to literature should be made […].

If a work is to be published in "Animals" it needs to be completely revised, it is not suitable for publication in its current state.

Author Response

Dear Reviewer, I would like to thank you for your very helpful advice, I have modified the manuscript following all your suggestions. I carried out a careful and thorough review of the manuscript and tried to make it clearer. I hope I have resolved all your doubts and that it is now suitable for publication. In addition, a native language reviewer has corrected the manuscript (certification letter download in June)

Round 2

Reviewer 1 Report

Comments to the authors

Thank you for your response and correction. I would like to add some corrections.

[Materials and methods]

Line 185-187: I think there is a typo.

[Results]

Abbreviation is used or not in the description. Please unify the description method. For example, “The shape of the femora head” (line 284) and “PFN and the FHC/DAR “(line 292).

[Discussion]

Line 326-334; The significance of the paper should be emphasized at the beginning of the discussion. Please describe the limitation in the later of the discussion.

Line 380; Please check if the FH/DAR is correct.

[Conclusions]

Line 457-459; There was no discussion of age in the paper. This study was evaluated at a median age of 13-15 months old dogs (range 12 months old -). Are dogs younger than 12 months old not indicated?

[Table]

Table 3; Please provide a description of the abbreviation of each breed (list the legend of table 4 in table 3).

Table5ï¼›I could not see it.

Author Response

Dear Reviewer,

thank you for the opportunity to revise the paper and re-submit it. I hope that the responses to your comments have helped improve the manuscript.

Reviewer 3 Report

Thank you to the authors for comprehensive answers. I am satisfied with their content.
I still think that radiologists should write about hip dysplasia in dogs.

Author Response

Dear Reviewer, I am glad that my efforts were appreciated and the goal of improving my study was achieved. I agree with you that more information needs to be provided regarding hip dysplasia; in fact this was a goal of my research. I hope that this study will be a suggestion for radiologists in the future to describe the development of hip dysplasia in dogs in greater detail.

Thank you very much.

Reviewer 4 Report

The article has been significantly improved. It can be printed in its current form in the journal "Animals".

Author Response

Dear Reviewer, I am glad that my efforts were appreciated and the goal was achieved.

David Andrew Goldsmith is a native English speaker and professor of English specialized in scientific translations. He carefully revised the manuscript.

Thank you very much.